# Clinical Characteristics and Outcomes of Patients with Cirrhosis Who Develop Infective Endocarditis

**DOI:** 10.3390/idr17020037

**Published:** 2025-04-21

**Authors:** Erika M. Dorff, Kyle Crooker, Torrance Teng, Tess Hickey, Max HoddWells, Ashwini Sarathy, Sean Muniz, Jennifer Lor, Amy Chang, Devika Singh, Jean Dejace, Elly Riser, Bradley J. Tompkins, Andrew J. Hale

**Affiliations:** 1Department of Medicine, University of Vermont Medical Center, Burlington, VT 05401, USA; erika.dorff@uvmhealth.org (E.M.D.);; 2Department of Medicine, Henry Ford Hospital, Detroit, MI 48202, USA; 3Larner College of Medicine at the University of Vermont, Burlington, VT 05401, USA; 4Department of Medicine, University of Utah Hospital, Salt Lake City, UT 84132, USA; 5Department of Medicine, Maine Medical Center, Portland, ME 04102, USA; 6Department of Emergency Medicine, University of Utah Hospital, Salt Lake City, UT 84132, USA; 7Department of Medicine, Rush University Medical Center, Chicago, IL 60612, USA; 8Department of Medicine, Highland Hospital, Oakland, CA 94602, USA

**Keywords:** infective endocarditis, cirrhosis

## Abstract

**Background:** Infective endocarditis (IE) is an increasingly common infection that results in significant morbidity and mortality. An important but under-analyzed subpopulation of patients with IE are those with concomitant cirrhosis. This study compared the characteristics and outcomes of patients with and without cirrhosis who were hospitalized with IE. **Methods:** The authors conducted a retrospective cohort study in adult patients with IE admitted at a single center from 2010 to 2020, comparing outcomes between those with and without cirrhosis at the time of admission. **Results:** A total of 22 patients with a history of cirrhosis and 356 patients without a history of cirrhosis were included. Over a quarter (27.3%) of those with cirrhosis experienced a decompensation event within two years of their admission for IE. Clinical features, microbiology, and direct complications from IE were largely similar between groups. There was no significant difference in IE-related mortality rates between groups, although, in an overall survival analysis, the group with cirrhosis did have a higher risk of all-cause mortality at 2 years (HR = 2.85; *p* = 0.012). **Conclusions:** This study highlights that IE in patients with cirrhosis may contribute to or trigger decompensation events. Further research is warranted to better understand morbidity outcomes in patients with cirrhosis who develop IE.

## 1. Introduction

Infective endocarditis (IE) is an increasingly common infection that results in significant morbidity and mortality [1,2]. Despite improvements in antimicrobials and surgical therapies, the 30-day and in-hospital mortality rate is around 10% and as many as one-third of patients who are hospitalized die within one year [1,3]. The annual incidence of IE has significantly increased in recent years in the United States, in part due to an increase in substance use disorder by injection, particularly among younger populations [4,5,6]. This increasing IE prevalence is also observed globally, which, combined with the rise in antimicrobial resistance, suggests that further research is needed on how best to manage IE [7,8]. People who inject drugs (PWIDs) are more likely to have high-risk comorbidities, such as concurrent HIV or hepatitis C virus (HCV) infection and cirrhosis or underlying liver disease [1,9]. Despite the known connection between IE, injection drug use, and concomitant cirrhosis, the outcomes of this subpopulation of patients are under-analyzed in the current literature.

Cirrhosis increases the risk of developing bacterial infections due to immune system dysfunction and systemic inflammation [10,11]. More specifically, cirrhosis can lead to the decreased clearance of bacteria and cytokines from circulation via the liver’s reticuloendothelial cells [12]. The development of these infections can also precipitate cirrhotic decompensation events, including hepatic encephalopathy and variceal bleeding, resulting in worse liver-related outcomes [13]. Patients with cirrhosis who develop sepsis are at risk of cytokine-induced apoptosis and necrosis, which can precipitate acute-on-chronic liver failure (ACLF) [12]. Bacteremia is more common in patients with cirrhosis, and mortality secondary to bacteremia in those with decompensated cirrhosis is much higher compared to those with compensated liver disease (70.0% vs. 30.0%, respectively) [7,14,15]. There has been a decrease in overall mortality rates with improvements in the early detection and antimicrobial prophylaxis of certain infections, particularly spontaneous bacterial peritonitis (SBP). However, mortality rates for bacteremia in patients with cirrhosis remain quite high, in part due to increased antimicrobial resistance and complications from multidrug-resistant (MDR) infections [7,15].

It has been reported that about 10% of patients with IE have a documented history of cirrhosis [12]. The clinical characteristics of IE, including the infection source and microbiology, in patients with cirrhosis have been described as similar to those of patients without chronic liver disease [16]. There have been some reports of IE following transjugular intrahepatic portosystemic shunt (TIPS) procedures, upper endoscopies, and liver biopsies; however, this is rarely described, and there are no specific recommendations regarding IE prevention in these procedures [16,17]. The prognosis of IE in patients with cirrhosis has been documented in the literature since 1961 [18]. Cirrhosis has been described as a notable prognostic risk factor for mortality in patients with IE, yet the relationship between these two processes is relatively unexplored [10]. In one study, the authors found that the rate of in-hospital mortality in IE was 14.0% for patients with compensated cirrhosis and 18.5% for those with decompensated cirrhosis [19]. Those with decompensated cirrhosis also demonstrated significantly higher rates of intensive care admissions and new-start dialysis and are less likely to receive surgical interventions compared to those without cirrhosis [19]. Moreover, with traditional risk factors such as viral hepatitis and heavy alcohol use remaining a global concern, coupled with increasing rates of metabolic dysfunction-associated steatotic liver disease (MASLD), cirrhosis is becoming more common, and it is expected that more patients with cirrhosis will develop infective endocarditis in the future [20].

While many of these studies discuss the risk of mortality in this population, few describe specific clinical and microbiologic differences between IE in patients with or without cirrhosis or morbidity outcomes, such as hepatic decompensation events. The goal of this study was to compare the clinical characteristics and outcomes of patients with and without cirrhosis who were hospitalized with infective endocarditis.

## 2. Methods

We conducted a retrospective cohort study of adult patients with IE admitted to the University of Vermont Medical Center (UVMMC) in Burlington, Vermont, from 2010 to 2020, comparing outcomes between those with or without cirrhosis at the time of IE admission. The UVMMC is a tertiary-care academic medical center with 620 licensed beds, serving urban and rural populations in Vermont and parts of upstate New York with a catchment area that covers approximately one million people. All patients over 18 years of age with IE, as identified by ICD-10 codes, were included. We excluded patients if they did not meet the definition of endocarditis as defined by the modified Duke criteria and IDSA guidelines. Two members of the study team performed a manual chart review of each patient. All reviewers received training, and differences were adjudicated by a third physician’s review. This study was approved by the University of Vermont Medical Center Institutional Review Board, under approval number STUDY00001869, on 3 December 2021.

We compared baseline demographic data, comorbidities, and clinical details between patients with IE. We further reviewed the patients’ histories to determine if they had a diagnosis of cirrhosis. A diagnosis of cirrhosis was based on clinical documentation of risk factors with imaging studies or liver biopsy results consistent with fibrosis, with or without evidence of decompensation as described below. The comorbidities compared included concurrent HIV, positive HBV serology, positive HCV serology, type 2 diabetes mellitus, chronic heart failure, and chronic kidney disease. We also compared the valve location and microbiologic profile, as well as valve replacement surgery. Our primary outcomes were 30-day, 90-day, 1-year, and 2-year mortality rates; IE-related mortality and IE-related readmission rates; and rates of decompensation events among patients with cirrhosis. Mortality was determined to be due to IE if the patient died as a result of hemodynamic collapse from septic shock, valvular complications, congestive heart failure, and/or other sequelae of IE, such as septic emboli. Hepatic decompensation events were defined as ascites, hepatic encephalopathy, variceal bleeding, or jaundice.

Data were input to a REDCap electronic data storage tool [21]. Statistical analyses were conducted with Stata (Stata 16.1, Stata Corp., LLC. College Station, TX, USA). Continuous variables were compared using the Wilcoxon rank-sum test, and categorical variables were compared using chi-square analysis or Fisher’s exact test. A significance level (α) of 0.05 was set prior to analysis, and any *p*-value less than 0.05 was determined to be statistically significant.

## 3. Results

A total of 378 patients with IE were included in this study based on the inclusion and exclusion criteria. Of these 378 patients, 22 were determined to have a cirrhosis diagnosis and the remaining 356 did not have a history of cirrhosis. Demographic data and risk factors are shown in Table 1. Parallel to the population demographics of this region, a majority of the study population identified as non-Hispanic white. The group with cirrhosis demonstrated significantly higher rates of HBV-positive serology (n = 2, *p* = 0.02) and diabetes (n = 10, *p* < 0.01). Additionally, there was a non-significant trend in the group with cirrhosis toward having a positive HCV serology (n = 10, *p* = 0.06) and more current alcohol use (n = 6, *p* = 0.14).

The characteristics of the patients with cirrhosis are shown in Table 2. The most common etiology of cirrhosis in these patients was alcohol use disorder (n = 10, 45.5%), followed by HCV (n = 8, 36.4%) and MASLD (n = 4, 18.2%). The mean MELD score on presentation was 18.6 (standard deviation: 8.4). Over a quarter of patients with cirrhosis were decompensated at the time of admission.

Comparisons of clinical characteristics for endocarditis are shown in Table 3. Admissions requiring ICU-level care were similar between the two cohorts. Patients without cirrhosis had a non-significantly higher rate of valve replacement surgery (n = 69, *p* = 0.40). The cohorts had similar valvular involvement, with the exception of pulmonic valve involvement being significantly more common in patients with cirrhosis (n = 2, *p* < 0.05). The microbiologic profiles of the cohorts were also similar, with MSSA being the most common bacteria for patients with and without cirrhosis.

Comparisons of outcomes are shown in Table 4. Patients with cirrhosis had higher overall mortality rates and a non-significant trend toward more deaths related to IE (n = 1, *p* = 0.49). Over a quarter of the patients with cirrhosis (27.8%) experienced a decompensation event within two years of their sentinel admission for IE. Sample sizes were adjusted for those who were not followed for the full 1-year or 2-year period following the date of admission, as noted in the footnote of Table 4.

Cox proportional hazards analysis was conducted for 2-year survival from the end of antimicrobial treatment to death for all patients that were observed for at least two years (n = 215). This model explains more variation in survival than a model that does not include cirrhosis as a factor (Prob > chi2 = 0.0143). Compared to patients without cirrhosis, patients with cirrhosis have a significantly higher hazard of death within 2 years of admission (HR = 4.21, *p* = 0.005). Figure 1 shows the associated Kaplan–Meier graph.

## 4. Discussion

Our study assessed the clinical features and outcomes of IE in patients with and without cirrhosis. Clinical features, microbiology, and direct complications from IE were largely similar between groups. Our study demonstrated similar IE-related mortality rates among those with and without cirrhosis, which was unexpected based on the prior literature, but could be due to the small sample size. The group with cirrhosis had a significantly higher hazard of death from any cause within 2 years of admission compared to the group without cirrhosis. The group with cirrhosis had significantly higher rates of HBV-positive serology, pulmonary valve involvement, and type 2 diabetes mellitus, the latter of which may increase the risk of bacterial infections in those with cirrhosis due to impaired immune system function [22].

This study also highlighted the possibility that IE in patients with cirrhosis may contribute to or trigger hepatic decompensation. While over a quarter of the cirrhosis cohort had experienced a hepatic decompensation event prior to presentation for IE, over a quarter of the cohort experienced a first or additional decompensation event within two years of IE admission. This is interesting in comparison to the previous literature demonstrating hepatic decompensation rates of 5–12% per year in patients with previously compensated cirrhosis [23]. Prior studies have illustrated that patients with cirrhosis who develop bacterial infections have a greater risk of developing decompensation compared to those without infections, with the 5-year incidence of decompensation being 45% vs. 15% in those with or without infections, respectively [24]. Portal hypertension is the primary driver behind hepatic decompensation, but bacterial infections remain one of the most common triggers of these events due to the activation of the systemic inflammatory response. Some of the most frequent infections include urinary tract infections and pneumonia. In turn, patients with decompensated cirrhosis are more susceptible to developing bacterial infections [24]. Hepatic decompensation events have been associated with higher mortality rates among patients with cirrhosis, particularly nosocomial infections [16]. Furthermore, a history of hepatic decompensation has been found to be an independent risk factor for in-hospital mortality in patients with IE [25]. Despite the known association between bacterial infections and hepatic decompensation, there is little to no data on how IE could drive the development of such events.

The prior literature has demonstrated that the presence of shock, acute kidney injury (AKI), and mechanical ventilation are predictors of mortality in patients with cirrhosis who develop IE [13]. Other studies suggest that higher rates of IE-related mortality among patients with cirrhosis are related to poor surgical candidacy [10]. Decompensated liver disease can result in an increased risk of bleeding due to coagulopathy and renal dysfunction, both of which contribute to perioperative mortality, especially in those undergoing cardiac surgery [26]. Furthermore, these patients are at an increased risk of complications as a result of inadequate source control, such as endogenous endophthalmitis [27]. In our study, the cirrhosis cohort similarly had a non-significant trend toward lower rates of valve replacement surgery, which is particularly concerning for these patients given their immune system dysfunction related to their liver disease [11]. This poses a challenge for those unable to obtain adequate source control with antimicrobials alone but considered too high risk to undergo surgical treatment. Alternative modalities of source control, such as transcatheter aspiration, could be considered in this subpopulation of patients, though data are limited as most of these therapies are relatively new [28]. Patients with cirrhosis who are considered to be poor surgical candidates, particularly those who are classified as Child–Pugh B or C, may benefit from the involvement of a multidisciplinary team to help determine the best treatment plan given their risk of surgical complications and high rates of mortality [24,29]. However, patients who are classified as Child–Pugh A may have a lower risk of surgical complications and perioperative mortality in IE [25].

There are limitations to this study. It was conducted at a single center and was retrospective. Our study population consisted of mostly white, non-Hispanic patients, which limits the generalizability of the findings. The sample size of the cirrhosis cohort was small, reducing the power of the study. Additionally, manual chart review is prone to ascertainment errors, leaving the possibility that some of the data collected are incomplete or were variably reviewed, and some patients may have been lost to follow-up with outcomes not captured. We mitigated this risk with multiple reviewers per patient. Moreover, patient deaths discovered on chart review were not verified on an independent death registry, so mortality data may be incomplete. Despite these limitations, this is one of the first studies to explore the rate of hepatic decompensation in patients with cirrhosis who develop IE.

While it has been shown that bacterial infections are associated with higher mortality rates in decompensated cirrhosis compared to compensated cirrhosis, little is known about how IE could contribute to decompensation events. Further research would be beneficial to elucidate the rates and types of hepatic decompensation events, in addition to morbidity outcomes, among these patients.

## 5. Conclusions

We found that patients with cirrhosis had significantly higher all-cause mortality compared to patients without cirrhosis, but specifically, IE-related mortality was similar between the two groups. However, this study highlights the possibility that IE could contribute to or trigger hepatic decompensation events. These results emphasize the need for additional research that focuses on both morbidity outcomes and ways to reduce IE-related mortality in patients with cirrhosis.

## Figures and Tables

**Figure 1 idr-17-00037-f001:**
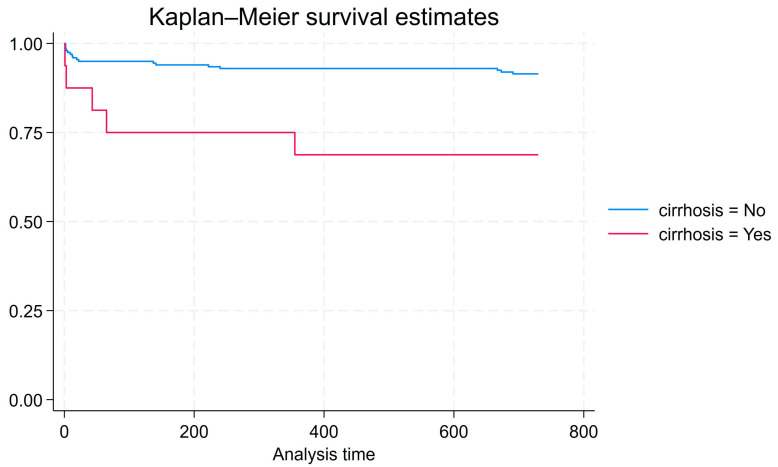
Kaplan–Meier graph for 2-year survival from end of antimicrobial treatment to death for all patients that were observed for at least two years (n = 215).

**Table 1 idr-17-00037-t001:** Demographic data and risk factors.

	No Cirrhosis (n = 356)		Cirrhosis (n = 22)		*p*-Value
	#	%	#	%	
Age, median (IQR)	56.8	(35.3–70.5)	58.5	(48.1–65.1)	0.66
Gender: female	140	(39.3)	10	(45.5)	0.57
Race: Caucasian	338	(96.6)	21	(95.5)	0.55
Ethnicity: non-Hispanic	343	(99.1)	22	(100.0)	1.00
Current smoker	139	(40.4)	9	(40.9)	0.96
Current alcohol use	54	(15.3)	6	(27.3)	0.14
Current IV drug use	120	(33.7)	10	(45.5)	0.26
Current substance use	117	(33.1)	8	(36.4)	0.77
Marijuana	47	(13.3)	4	(18.2)	0.52
Methamphetamines	20	(5.7)	0	(0.0)	0.62
Cocaine	62	(17.6)	7	(31.8)	0.09
Opioids	56	(15.9)	2	(9.1)	0.55
Other	8	(2.3)	2	(9.1)	0.11
Chronic kidney disease	69	(19.4)	7	(33.3)	0.12
HIV ^a^	2	(0.7)	1	(5.6)	0.16
HBV ^b^	2	(0.7)	2	(10.5)	0.02
HCV ^c^, positive serology	93	(31.4)	10	(52.6)	0.06
Pre-existing prosthetic heart valve	71	(19.9)	3	(13.6)	0.59
Indwelling cardiac device	37	(10.4)	3	(13.6)	0.72
Chronic heart failure	52	(14.6)	2	(9.1)	0.75
Diabetes ^d^	73	(20.5)	10	(45.5)	<0.01

^a^ HIV = human immunodeficiency virus; ^b^ HBV = hepatitis B virus; ^c^ HCV = hepatitis C virus; ^d^ includes both type 1 and type 2 diabetes mellitus without differentiation between controlled or uncontrolled.

**Table 2 idr-17-00037-t002:** Characteristics of patients with cirrhosis.

Cause of Cirrhosis, n (%)	Number (%), n = 22
Alcohol use disorder	10 (45.5)
MASLD ^a^	4 (18.2)
HCV ^b^	8 (36.4)
HBV ^c^	2 (9.1)
Other: Wilson’s disease, alpha-one antitrypsin deficiency, hemochromatosis, autoimmune	3 (13.6)
MELD ^d^ score on presentation, median (IQR)	18 (11–23)
Compensated at time of presentation	16 (72.7)
Decompensated (ascites, variceal bleeding, other cirrhosis-related bleeding, hepatic encephalopathy, spontaneous bacterial peritonitis) at time of presentation	6 (27.3)

^a^ MASLD = metabolic dysfunction-associated steatotic liver disease; ^b^ HCV = hepatitis C virus; ^c^ HBV = hepatitis B virus; ^d^ MELD = Model for End-Stage Liver Disease.

**Table 3 idr-17-00037-t003:** Clinical characteristics of the endocarditis episode.

	No Cirrhosis		Cirrhosis		*p*-Value
Admission required ICU, n (%)	150	(42.1)	9	(40.9)	0.91
Pitt bacteremia score, median (IQR)	1	(0–2)	1	(0–2)	0.89
Length of hospitalization, median (IQR)	15	(7–30)	13.5	(10–32)	0.85
Heart valve location for endocarditis, # (%)					
Tricuspid	85	(23.9)	5	(22.7)	0.90
Pulmonic	2	(0.6)	2	(9.1)	0.02
Aortic	98	(27.5)	8	(36.4)	0.37
Mitral	100	(28.1)	6	(27.3)	0.93
Unknown	53	(14.9)	4	(18.2)	0.76
Valve replacement surgery, n (%)	69	(19.4)	2	(9.1)	0.40
Microbiology					
MSSA ^a^	119	(33.4)	9	(40.9)	0.47
MRSA ^b^	50	(14.0)	2	(9.1)	0.75
*Streptococcus viridans*	32	(9.0)	0	(0.0)	0.24
Other *Streptococcus* species	63	(17.7)	1	(4.6)	0.15
*Enterococcus faecalis*	23	(6.5)	1	(4.6)	1.00
*Enterococcus faecium*	1	(0.3)	0	(0.0)	1.00
HACEK ^c^ group	7	(2.0)	0	(0.0)	1.00
*Candida*	4	(1.1)	0	(0.0)	1.00
Other	54	(15.2)	9	(40.9)	<0.01

Days of positive blood cultures, median (IQR)	2	(1–3)	1	(1–4)	0.90
IV antimicrobial duration in days, median (IQR)	42	(21–42)	28	(28–44)	0.33
Total days of antimicrobials, median (IQR)	42	(29–44)	42	(32–46)	0.45

^a^ MSSA = methicillin-susceptible Staphylococcus aureus; ^b^ MRSA = methicillin-resistant Staphylococcus aureus; ^c^ HACEK = Haemophilus, Aggregatibacter, Cardiobacterium, Eikenella, Kingella.

**Table 4 idr-17-00037-t004:** Comparisons of outcomes.

	No Cirrhosis (n = 356)		Cirrhosis (n = 22)		*p*-Value
	#	%	#	%	
30-day mortality rate	10	(2.8)	1	(4.6)	0.49
90-day mortality rate	18	(5.1)	2	(9.1)	0.33
1-year mortality rate ^a^	20	(6.7)	3	(15.0)	0.17
2-year mortality rate ^a^	26	(7.4)	3	(16.7)	0.17
Death related to endocarditis	10	(2.8)	1	(4.6)	0.49
30-day endocarditis mortality rate	8	(2.3)	1	(4.6)	0.42
90-day endocarditis mortality rate	9	(2.6)	1	(4.6)	0.46
1-year endocarditis mortality rate ^b^	9	(3.1)	1	(5.0)	0.46
2-year endocarditis mortality rate ^b^	9	(4.2)	1	(5.6)	0.39
Rate of readmission					
30 days	51	(14.3)	0	(0.0)	0.06
90 days	69	(19.4)	2	(9.1)	0.23
1 year ^a^	60	(27.9)	5	(27.8)	0.99
2 years ^a^	68	(31.6)	6	(33.3)	0.88
Patients with cirrhosis with decompensation event within 2 years of infective endocarditis ^c^			5	(27.8)	

^a^ No cirrhosis—n = 215; cirrhosis—n = 18. ^b^ No cirrhosis—n = 212; cirrhosis—n = 18. ^c^ n = 18.

## Data Availability

The study data is available upon reasonable request to the authors.

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
