# Peer review of "Clinical Characteristics and Outcomes of Patients with Cirrhosis Who Develop Infective Endocarditis"

_2036-7449, 2025, doi:10.3390/idr17020037_

Round 1

Reviewer 1 Report (Previous Reviewer 2)

Comments and Suggestions for Authors

The manuscript was improved. Thank you.

Author Response

We thank the reviewer for their time and prior suggestions to improve our manuscript. 

Reviewer 2 Report (Previous Reviewer 3)

Comments and Suggestions for Authors

In this version, the authors provided reasonable revisions in response to feedback from myself and other reviewers.

Only one minor point I noticed is the conclusion "We found that rates of IE-related mortality were similar among patients with or without cirrhosis."

In the results section, the authors focused more about all-cause mortality rather than IE-related mortality (Cox regression analysis and Kaplan-Meier curve for IE-related mortality was not reported). It seems odd to me.

I would recommend revising the conclusion sentence to "We found patients with cirrhosis had a significant all-cause mortality compared to patients without cirrhosis, but IE-related mortality was similar between two groups." or something similar. And would recommend adding Cox-regression analysis and K-M curve for IE-related mortality.

Other than that, I have no additional comments.

Author Response

We thank the reviewer for their time and prior suggestions to improve our manuscript.

We thank the reviewer for their comment. We have discussed this with our statistician. We agree that this is a good idea to include. Unfortunately, we do not believe we have enough IE-related mortality to create a reasonable K-M curve. Our statistician felt we could not produce a robust K-M curve given only 1 IE-related death in the cirrhosis cohort. We apologize.

We did amend the sentence in the conclusion section as requested by the reviewer.

Reviewer 3 Report (Previous Reviewer 4)

Comments and Suggestions for Authors

I've carefully read this revised version of the paper.

Even if improved, the manuscript still has important limitation that make it not suitable for publication. In particular, the Authors did not resolve the queries highlighted in the first reviews, especially in the field of sample size and generalizability.

Author Response

We thank the reviewer for their comment. We hope that the additional edits we have made in this round of review helped strengthen the paper and make it suitable for publication.

While it is true we have a low number of patients in the  cirrhosis cohort (n = 22), the total sample size of the study (n = 378, captured over a 10 year period) is fairly robust and comparable or larger to similar data sets. Thus we do feel the sample size, while certainly a limitation, does not disqualify the value of the findings.

While our population was homogenous in certain ways, for instance it was predominantly Caucasian, we acknowledge this readily in the limitations. Race and ethnicity are not clearly, as established by prior literature, independent risk factors for worse outcomes in endocarditis or cirrhosis, so while we agree this is a limitation, we do disagree it disqualifies any generalizability of our findings.

Within the limitations of the study, we do believe the findings are robust, add to the current literature, and would be worth sharing. 

Reviewer 4 Report (New Reviewer)

Comments and Suggestions for Authors

Dear authors,

I have read your manuscript carefully and with interest. I do think that this study provides a valuable analysis of 22 cirrhotic patients with endocarditis compared to 356 patients in a control group. While the findings are important, the small sample size limits generalizability but you do acknowledge that in your paper. Despite this limitation, the study represents a step forward in understanding the intersection of cirrhosis and endocarditis. Here are my major comments for your consideration in order improve the manuscript. 

  1. It is well-known that infections, especially severe ones like infective endocarditis (IE), can trigger acute-on-chronic liver failure in cirrhotic patients, often signaling liver disease decompensation. This connection should be emphasized as it helps contextualize the findings within the broader understanding of cirrhosis.
  2. The introduction could benefit from a more detailed explanation of immune compromise in cirrhosis, which would help readers understand why cirrhotic patients are particularly vulnerable to severe infections like IE. Including a review of the literature on this subject and addressing the role of non-alcoholic fatty liver disease (NAFLD), now called metabolic-associated fatty liver disease (MASLD), would strengthen the background. Additionally, as cirrhosis progresses, Kupffer cell damage increases susceptibility to infections, which is an important point to highlight.
  3. A global perspective on the epidemiology of IE and bacteremia in cirrhotic patients would also strengthen the introduction, especially considering the growing challenge of antimicrobial resistance in resource limited countries. Including references such as "Bacteremia in patients with liver cirrhosis in the era of increasing antimicrobial resistance" would broaden the context of the findings.
  4. The methodology is clearly outlined and well-executed with no major criticisms.
  5. In the results section, Table 1, titled "Baseline Characteristics," should be renamed "Demographic Data and Risk Factors" to better reflect its content. Additionally, while diabetes appears statistically significant, it is unclear whether this refers to type I or II diabetes or whether it was controlled or uncontrolled. Clarifying this would aid in interpreting the results.
  6. In the discussion, comparing pathogens responsible for bacteremia and IE in your cohort with those found in developing countries would be useful. This comparison would highlight geographical differences in infection patterns and emphasize the challenges posed by antimicrobial resistance in resource-limited settings.

Author Response

We thank the reviewer for their comment.

We thank the reviewer for their suggestion. We have included further information in the introduction section, specifically lines 139-141, regarding the development of ACLF in patients with severe infections.

Thank you for your comment. We have provided a more detailed explanation of the immune compromise that is present in cirrhosis, specifically lines 135-137 and lines 139-141. Additionally, we have addressed the increasing burden of MASLD in lines 163-167 of the introduction.

Thank you for providing us with this article. We agree that antimicrobial resistance is increasing in both the US and globally and have commented on this in lines 129-130. We have also cited the article that was provided in lines 129-130 as well as 148-149.

We thank the reviewer for their comment.

We agree and have updated the results section in lines 243-244 as well as the title of Table 1 to reflect this change. We also provided further clarification regarding diabetes in the footnote of Table 1. We included type I and type II diabetes mellitus and did not differentiate between controlled or uncontrolled.

We agree with the reviewer that it would be useful to compare pathogens found in cohorts in developing countries. However, we have found that such data is sparse and believe that more research is warranted to better support this comparison.

This manuscript is a resubmission of an earlier submission. The following is a list of the peer review reports and author responses from that submission.

Round 1

Reviewer 1 Report

Comments and Suggestions for Authors

Thank you for inviting me to review this manuscript. It is interesting and well written. I have some comments that might be of use:

1.     Line 7: change ‘Unviserity’ to university

2.     Based on the information for authors, I suppose that all that detailed description in the affiliations is not needed. Please change that

3.     Line 46: A total of 22 patients with a history of cirrhosis and 356 patients without a history of cirrhosis were included  This is a result, and does not belong to the discussion section

4.     Lines 51 and 52 are not that clear. Please consider rephrasing them

5.     Line 90: ‘granular’? What does it mean here?

6.     Line 105: The protocol number and the date of the ethics approval should be provided

7.     Statistical analysis in the methods section: The level of significance (a) should be provided

8.     Table 2: The causes of cirrhosis could be stated in a decreasing frequency. It will slightly improve readability of the table

9.     Table 2: I am not sure that presenting the MELD score with a mean and SD is relevant, since this is not a continuous variable. It is discrete (no patient can have a score of 18.6). Presenting the median and the IQR is more appropriate

10.  All tables: footnotes should be added under every table explaining all abbreviations (MELD, MSSA, HIV, etc)

11.  Table 3: Replace Strep with Streptococcus

12.  Table 3: similar for duration of hospitalization, days of positive blood cultures, IV antibiotic duration in days, and total days of antibiotics as in comment no 9

13.  I am very curious to see what ‘other’ microorganisms are isolated, especially in patients with cirrhosis. There may be an association of specific pathogens with cirrhosis in these patients

14.  Replace antibiotics with antimicrobials throughout the text. Antibiotics is a term mostly used for bacteria – the authors showed that some Candida species were isolated as well

15.  The discussion is relatively poor and should be expanded. The authors are encouraged to either synthesize the existing evidence in the literature regarding patients with endocarditis and cirrhosis and compare their findings, or directly compare their findings with reviews on patients with infective endocarditis and cirrhosis that have already been published

16.  The authors have not commented on the higher occurrence of IE in the pulmonary valve. I did not see that in the results or the discussion either

Reviewer 2 Report

Comments and Suggestions for Authors

Authors demonstrated that there was no significant difference in IE-related mortality rates between those with and without cirrhosis, although in overall survival analysis the group with cirrhosis did have higher risk of mortality 2 years (HR=2.85; p=0.012).

1.    “In one study, the authors found the rate of in-hospital mortality of IE was 14.0% for patients with compensated cirrhosis and 18.5% for decompensated cirrhosis.” Authors should indicate the references.

2.    Why did authors select these 22 patients with a history of cirrhosis and 356 patients without a history of cirrhosis? Authors should describe the methods in more details.

3.    Authors should describe and discuss the reason for the selection of patients with cirrhosis more.

4.    Authors should discuss more about the association between IE and cirrhosis. See the following references: Shah MK, et al. Eye (Lond). 2025 Jan;39(1):125-132. doi: 10.1038/s41433-024-03390-w. PMID: 39402169; Van Hemelrijck M, et al. Front Cardiovasc Med. 2023 Aug 24;10:1223878. doi: 10.3389/fcvm.2023.1223878. PMID: 37692048….

5.    How the symptoms, diagnosis, treatment and prognosis of the 22 patients? Authors should demonstrate them in one table.

6.    Pelaquier et al. reported that the use of aminosides and rifamycin should be reassessed in LC, and cardiac surgery should be considered for selected patients (Eur J Gastroenterol Hepatol. 2018 Oct;30(10):1216-1223. doi: 10.1097/MEG.0000000000001155. PMID: 29727379). Authors should mention about the prophylaxis.

Reviewer 3 Report

Comments and Suggestions for Authors

This is a single center retrospective study to assess the characteristics and outcomes of cirrhotic patients with IE and non-cirrhotic patients with IE. Their cohort included 356 patients with IE but without cirrhosis and 22 cirrhotic patients with IE. Cirrhotic patients had higher rate of HBV positivity and diabetes compared to those without cirrhosis. There was not a significant difference in IE characteristics. Cirrhotic patients had more all-cause mortality in 2 years and 27.8% of cirrhotic patients had a decompensation event within 2 years of IE event.

Although I do not see a direct association between IE and cirrhosis except hepatitis C due to IV drug use, or intravascular device placement such as TIPS for cirrhosis, which are rare events, I still feel this study is an interesting one to focus on these two clinical entities. Due to limited numbers of cirrhotic IE patients, the analyses had been underpowered, but this study would be a good one to lead to future larger studies.

I have several suggestions and questions for this study as below.

1.       Methods – Some of the variables should be clearly defined. For example, how cirrhosis was defined? In addition, how the reviewers attributed mortality due to endocarditis? These definitions should be clearly mentioned in the manuscript or summarized in a supplemental table.

2.       Methods – the authors should include what kind of statistical analyses were conducted in the methods section.

3.       Table 3. Microbiology – what organisms were included in “other” category? I think it should be included as a footnote.

4.       Table 4. Mortality rate, endocarditis mortality rate and rate of readmission – why they do not add up? I think it should be cumulative (like 90-day mortality should include all who died within 90 days from index day, including those who died within 30 days)

5.       Results – I do not think use of Cox proportional hazard was appropriate because it violated the proportional hazard assumption (Kaplan-Meier curve crossed before day 50). I suggest either using a different start time (for example, use day 42 – when antibiotic treatment ends – to predict long-term outcome of patients who survived acute phase of IE), or including an interaction between a predictor and time to calculate the time-dependent hazard ratio.

6.       Discussion, lines 164-165. “Our study demonstrated similar IE-related mortality rates among those with and without cirrhosis, which was unexpected based on prior literature.” – I do not think the authors can say anything definitive as there would be a concern for type 2 error because of small number of cirrhosis patients. I would suggest revising that sentence.

7.       Discussion, lines 171-175. The authors compared % of decompensation event to a historical hepatic decompensation rate. I do not think it a fair comparison because 27.3% of cirrhotic patients already got a decompensation event at the time of presentation, and there was no adjustment in the characteristics in patients of this cohort and historical cohort. I would recommend removing this part or adding a nuanced sentence to explain the possible difference in the baseline characteristics.

8.       Discussion, lines 181-182. “The most frequent infections include urinary tract infections, spontaneous bacterial peritonitis, and pneumonia.” The sentence may be accurate but as the authors define SBP as a decompensation event, it was confusing to deal SBP as a trigger for decompensation or event itself. I would suggest revising the sentence to make it clear.  

Reviewer 4 Report

Comments and Suggestions for Authors

The Authors described impact of cirrhosis on the outcome of infective endocarditis. Despite the study population is quite big, the cirrhosis group, unfortunately, consists only of 22 patients.

The paper is well written, data are adequately presented as the introduction.

All the conclusions and considerations of the Author's, on the other hand, were strongly influenced by the cirrhotic group sample size. IE: it is very difficult to generalize the consideration of an higher incidence of hbv infection with only 2 cases (out ult 2). This data is inevitably influenced by the sample size and with scarse statistical significance. 

Moreover, 2 years mortality seems to be difficult to be related to a previous EI event without a clear description of oatients' history in these 2 years (other infection or clinically significant event may occured in such a long period)

Reviewer 5 Report

Comments and Suggestions for Authors

Title: Clinical characteristics and outcomes of patients with cirrhosis who develop infective endocarditis.

Reviewer Summary

Infective endocarditis (IE) is a common infection that results in considerable amount of morbidity and mortality. Concomitant cirrhosis patients are a significant however little-studied subgroup of IE patients. In this study authors tried to compare characteristics and outcomes of patients with and without cirrhosis who were hospitalized with IE. This is a retrospective cohort study done in adult patients. These patients have IE when they are admitted. Authors compared outcomes between those with and without cirrhosis. 22 patients had cirrhosis, and 356 patients had no cirrhosis. Authors found that over a quarter of those with cirrhosis experienced a decompensation event within two years of their admissions for IE. Clinical features, microbiology, and direct complications from IE were almost similar between groups. There was no significant difference in IE related mortality between groups. However, group with cirrhosis did have higher risk of mortality 2 years. In conclusion infective endocarditis in patients with cirrhosis may contribute to or trigger decompensation events.

Weaknesses:

1.     Retrospective studies are valuable tool for studying rare diseases, however these studies have several drawbacks including bias, reproducibility, missing data and need large sample size. 

2.    Authors conducted this study in mostly in white people. That means we can not generalize the results to diverse population.  

3.    These types of study need large sample size. People with cirrhosis only 22 people, and with no cirrhosis, 356 people. It’s difficult to compare the data when there is a large variation in sample size. 

4.    Authors need to explain more clearly why the combination of IE and cirrhosis is more concerning.

5.    Authors need to clearly explain or point out about the early diagnosis, aggressive antibiotic therapy, multidisciplinary approach to treat this condition and liver transplant. 

Round 2

Reviewer 1 Report

Comments and Suggestions for Authors

The manuscript has been improved.

Author Response

Thank you for your time and suggestions. We appreciate your review and feedback.

Reviewer 2 Report

Comments and Suggestions for Authors

Authors should describe the reasons why authors selected these 22 patients with a history of cirrhosis and 356 patients without a history of cirrhosis, in introduction and more in the methods section.

Author Response

Please check the file in the attachment.

Reviewer 3 Report

Comments and Suggestions for Authors

There are a couple of point which need to be revised.

  1. Explanation was not provided how the authors attributed mortality due to endocarditis.
  2. Use of Cox proportional regression analysis. It is inappropriate to use this analysis if it obviously violates proportional hazard assumption. It should be consulted to a statistician to choose the appropriate analytical method. 

Author Response

(The authors gave the same response as above.)

Reviewer 4 Report

Comments and Suggestions for Authors

I have carefully read the responses but, unfortunately, the Authors did not provide any improvements for this manuscript. 

As a consequence it cannot be accepted, according to me, in the present form.

Author Response

(The authors gave the same response as above.)
